# Shape Sensing of Plate Structures Using the Inverse Finite Element Method: Investigation of Efficient Strain–Sensor Patterns

**DOI:** 10.3390/s20247049

**Published:** 2020-12-09

**Authors:** Rinto Roy, Alexander Tessler, Cecilia Surace, Marco Gherlone

**Affiliations:** 1Department of Mechanical and Aerospace Engineering, Politecnico di Torino, Corso Duca degli Abruzzi 24, 10129 Torino, Italy; rinto.roy@polito.it (R.R.); marco.gherlone@polito.it (M.G.); 2Structural Mechanics and Concepts Branch, NASA Langley Research Center, Mail Stop 190, Hampton, VA 23681-2199, USA; alexander.tessler-1@nasa.gov; 3Department of Structural, Geotechnical and Building Engineering, Politecnico di Torino, Corso Duca degli Abruzzi 24, 10129 Torino, Italy

**Keywords:** shape sensing, inverse Finite Element Method, structural health monitoring, plate structures, strain sensors

## Abstract

Methods for real-time reconstruction of structural displacements using measured strain data is an area of active research due to its potential application for Structural Health Monitoring (SHM) and morphing structure control. The inverse Finite Element Method (iFEM) has been shown to be well suited for the full-field reconstruction of displacements, strains, and stresses of structures instrumented with discrete or continuous strain sensors. In practical applications, where the available number of sensors may be limited, the number and sensor positions constitute the key parameters. Understanding changes in the reconstruction quality with respect to sensor position is generally difficult and is the aim of the present work. This paper attempts to supplement the current iFEM modeling knowledge through a rigorous evaluation of several strain–sensor patterns for shape sensing of a rectangular plate. Line plots along various sections of the plate are used to assess the reconstruction quality near and far away from strain sensors, and the nodal displacements are studied as the sensor density increases. The numerical results clearly demonstrate the effectiveness of the strain sensors distributed along the plate boundary for reconstructing relatively simple displacement patterns, and highlight the potential of cross-diagonal strain–sensor patterns to improve the displacement reconstruction of more complex deformation patterns.

## 1. Introduction

In the past several decades, Structural Health Monitoring (SHM) has been the subject of intense research for application to civil, aerospace, and naval structures. SHM promises the possibility of evaluating the structural integrity under real-time operational conditions. The SHM data can lead to improvements in design, failure predictions, safety, and more effective maintenance strategies [1]. In the framework of SHM, shape sensing, i.e., the reconstruction of the displacement field of a structure from discrete surface strains, represents an enabling technology. Shape sensing can not only provide an effective shape control strategy for those structures that carry antennas [2] but is also an important tool to provide feedback to smart structures with morphing capabilities [3]. Moreover, the reconstruction of the displacement field can be the first step towards the evaluation of strains and stresses across the entire or partial structural regions [4,5].

The interest in shape-sensing methodologies has also grown along with the introduction of new strain measurement systems. Traditional strain gages are now accompanied by strain sensors based on fiber optics that are more robust with respect to electromagnetic interferences can be attached and/or embedded, and provide a much higher spatial density of measurement points [6,7,8].

The current shape sensing methodologies can be divided into four main categories [9]: integration of experimental strains; use of basis functions to approximate the displacement field; use of Neural Network (NN); and use of a FEM-discretized variational principle.

Approaches that obtain the displacement field by integrating strains measured at discrete locations have been mainly applied to beam-like structures [6,10], and only a few papers presented an extension to two-dimensional cases [11]. In [6], Ko et al. proposed a one-dimensional scheme based on classical beam theory to evaluate the deflection and bending rotation of a beam at the same locations where strains have been measured. Using a two-line strain-sensing strategy, the cross-sectional twist angle can be also reconstructed, as demonstrated on an aircraft wing [6].

There are a number of shape-sensing approaches that make use of an a priori choice of basis functions and of a set of corresponding unknown weights to approximate the strain field [12,13,14,15,16,17,18]. Then, the displacement field is obtained by means of strain–displacement relationships, where typically the classical Bernoulli–Euler beam or the Kirchhoff plate assumptions are used. Basis functions can be global or piece-wise polynomials [12,13] or the structural mode shapes [14,15,16,17,18]. The latter approaches are grouped under the name of Modal Methods and can be based on experimental mode shapes [14,15], analytical mode shapes [16], or mode shapes evaluated through a FE model [17,18]. The experimental evaluation of the mode shapes can be onerous; on the other hand, it does not require any knowledge of the material properties. Another key aspect of these methodologies is the choice of the mode shapes to be included in the analysis and the related risk of inaccuracies when high-frequency excitations are present and only low-order modes are available [18].

A few efforts have focused on the use of neural networks to the displacement–field reconstruction [19,20]. The main drawback of this methodology is that its accuracy is strongly dependent on the choice of the load cases used for the training.

The inverse Finite Element Method (iFEM) is a variationally based approach to shape sensing, where an error functional is discretized using the finite element framework. Pioneering works in this field have been authored by Tessler and Spangler [21,22]. The key idea is to employ a weighted-least-square functional that is discretized by means of finite elements. The functional represents the least-square error between the experimentally measured strains and those corresponding to the finite element approximations. Minimization of the error functional leads to a system of algebraic equations whose unknowns are the nodal degrees of freedom of the finite element mesh. Once the nodal unknowns are obtained, the displacement field can be fully reconstructed using the corresponding shape functions. The initial efforts in this field dealt with thin-walled structures [21,22], where a three-node inverse shell element (iMIN3) was developed on the basis of Mindlin (first-order Shear Deformation) theory [23]. A four-node inverse shell element was derived by Kefal et al. [24], who also developed an inverse curved shell-element [25], and used an iso-geometric approach to iFEM [26]. Gherlone et al. [27,28] applied three-dimensional (3D) beam approximations within iFEM to study shape sensing in truss, beam, and frame structures instrumented with strain gauges. Moreover, two-dimensional inverse elements for the shape-sensing of multilayered composite and sandwich plates have been recently proposed in [4,5,29]. Several iFEM applications have been presented, ranging from marine structures [25,30,31] to composite wing boxes [32], using numerically generated strain data [4,5,24,25,26,27,32] and experimentally measured strains [9,28]. Since iFEM employs only strain–displacement relations, structures under either static or dynamic loadings can be analyzed without any a priori knowledge of the material, inertial, loading, or damping properties of the structure [22,27]. The iFEM solution method is very efficient. Even under dynamic loading, when strains are time dependent, the coefficients matrix of the method is inverted only once [22], whereas the right-hand side vector is updated according to changes in the strain values. Key features of iFEM are represented by the availability of strain sensors and by sensor locations. The first topic has been addressed (in [33] and subsequently in most of the iFEM papers) by using penalization strategies for the inverse elements without strain data. The effect of sensor locations has also been investigated in [24] for a thin-walled cylinder, in [30] for a chemical tanker, in [31] for a Panamax container ship, in [29] for a wing-shaped sandwich plate, and in [32] for a composite wing-box. In [29], both the strain rosettes and FBG sensors are considered for the strain measurement. In [32], genetic algorithms are used to obtain the optimal sensor locations for an accurate displacement field reconstruction of a wing box under bending and torsional loading.

Although previous research efforts [29,32] proposed sensor configurations that offered relatively high reconstruction accuracy, the problems considered were structures undergoing relatively simple bending or torsion deformations, where the displacement field was not complex. Additionally, the investigation of the reconstructed displacements was restricted to certain points or sections at the tip or edge of the structure. These studies have not adequately examined the reconstruction quality at locations far from the strain sensors. The use of simple iterative or optimization procedures for sensor placement leads to the choice of sensor configurations, which appear random at first inspection, and it is difficult to offer any physical reasoning to justify their selection. This leads to difficulties when extending the method to more complex structures. Overcoming some of these deficiencies is the focus of the current work.

This paper aims to establish certain elementary sensor configurations that can produce accurate iFEM shape sensing predictions in a systematic manner. Herein, we focus on a simple rectangular plate and develop a greater understanding of how the choice of sensor locations affects the accuracy of each reconstructed displacement component. The sensor configurations chosen are based on simple patterns, commonly observed in load-bearing frame structures, and can easily be reproduced in a laboratory setting. Both simple and complex displacement fields are explored, and the reconstructed displacement field is subject to a rigorous investigation at locations near and far from a sensor array. This helps quantify the reconstruction accuracy over the entire structure. Additionally, the effect of the inverse element used is also explored by comparing the reconstruction results using two different inverse elements. This helps to avoid any bias in sensor placement related to the selection of any specific element. The understanding garnered from such a rigorous investigation on a simple structural domain will inspire and offer a pathway for sensor placement strategies in more complex structures.

The paper is organized as follows. The iFEM theoretical formulation is presented in Section 2, along with a description of the two inverse finite elements. The proposed sensor configurations are described in Section 3. The shape-sensing performance of each sensor configuration is explored on a cantilevered rectangular plate subject to low and moderately high vibrational modes. Displacement reconstruction results for various mode shapes are discussed, and the accuracy of the two inverse elements is compared. Finally, conclusions and future work opportunities are presented in Section 4.

## 2. Theoretical Foundation of the Inverse Finite Element Method

The iFEM formulation for plate and shell structures is based on the displacement assumptions of the Mindlin plate theory [34]. The theory employs five kinematic variables: $u,\;v,\;w,\;\theta_{x},$ and $\theta_{y},$ associated with the mid-plane of the plate/shell. The Cartesian components of the displacement vector can be expressed as
(1)$$u_{x}(x,y,z)=u+z\theta_{y},\;u_{y}(x,y,z)=v-z\theta_{x},\;u_{z}(x,y,z)=w,$$
where u and v denote the mid-plane displacements along the x and y axes, $\theta_{x}$ and $\theta_{y}$ are the bending rotations about the x and y axes, and w represents the average transverse displacement of the shell along the z-axis (see Figure 1).

The strain field can be obtained by differentiating the displacement field of Equation (1), resulting in the linear strain displacement relations
(2)$$\left\{\begin{array}{l} \varepsilon_{xx}\\ \varepsilon_{yy}\\ \gamma_{xy} \end{array}\right\}=\left\{\begin{array}{l} \varepsilon_{x0}\\ \varepsilon_{y0}\\ \gamma_{xy0} \end{array}\right\}+z\left\{\begin{array}{l} \kappa_{x0}\\ \kappa_{y0}\\ \kappa_{xy0} \end{array}\right\}\equiv e(u)+zk(u).$$

The strain field is influenced by two major contributions: one pertaining to the in-plane stretching of the mid-plane, referred to as the membrane strain measure e(u), and another related to the bending of the mid-plane, referred to as the curvature vector or the bending strain measure k(u). These membrane and bending strain measures are given in terms of the kinematic variables as
(3)$$e(u)=\left\{\begin{array}{l} \varepsilon_{x0}\\ \varepsilon_{y0}\\ \gamma_{xy0} \end{array}\right\}=\left[\begin{array}{ccccc} \frac{\partial }{\partial x} & 0 & 0 & 0 & 0\\ 0 & \frac{\partial }{\partial y} & 0 & 0 & 0\\ \frac{\partial }{\partial y} & \frac{\partial }{\partial x} & 0 & 0 & 0 \end{array}\right]\left\{\begin{array}{l} u\\ v\\ w\\ \theta_{x}\\ \theta_{y} \end{array}\right\},\;k(u)=\left\{\begin{array}{l} \kappa_{x0}\\ \kappa_{y0}\\ \kappa_{xy0} \end{array}\right\}=\left[\begin{array}{ccccc} 0 & 0 & 0 & 0 & \frac{\partial }{\partial x}\\ 0 & 0 & 0 & -\frac{\partial }{\partial y} & 0\\ 0 & 0 & 0 & -\frac{\partial }{\partial x} & \frac{\partial }{\partial y} \end{array}\right]\left\{\begin{array}{l} u\\ v\\ w\\ \theta_{x}\\ \theta_{y} \end{array}\right\}.$$

Mindlin plate theory also accounts for transverse shear deformation, giving rise to uniform transverse shear strains across the thickness, i.e.,
(4)$$g(u)=\left\{\begin{array}{l} \gamma_{xz0}\\ \gamma_{yz0} \end{array}\right\}=\left[\begin{array}{ccccc} 0 & 0 & \frac{\partial }{\partial x} & 0 & 1\\ 0 & 0 & \frac{\partial }{\partial y} & -1 & 0 \end{array}\right]\left\{\begin{array}{l} u\\ v\\ w\\ \theta_{x}\\ \theta_{y} \end{array}\right\}.$$

For each inverse element, e, the element degrees-of-freedom (DOF) vector, $u^{e}$, can be defined as
(5)$$u^{e}={\left[u_{1}^{e}\;u_{2}^{e}\;\ldots \;u_{n}^{e}\right]}^{T},$$
where $u_{i}^{e}$ denotes the DOF of each node of the element. The element displacement vector can be used to write the strain–displacement relations in terms of the nodal DOF of the element
(6)$$\left\{\begin{array}{l} \varepsilon_{xx}\\ \varepsilon_{yy}\\ \gamma_{xy} \end{array}\right\}\equiv e(u^{e})+zk(u^{e})=B^{m}u^{e}+zB^{b}u^{e},\;\left\{\begin{array}{l} \gamma_{xz}\\ \gamma_{yz} \end{array}\right\}\equiv g(u^{e})=B^{s}u^{e},$$
where $B^{m},\;B^{b}$, and $B^{s}$ are matrices of shape function derivatives corresponding to the membrane, bending, and transverse shear strains.

Herein, three and four-node inverse elements are discussed. The three-node triangular element, referred to as iMIN3 [23], is a constant strain element with five DOF at each node,
(7)$$u_{i}^{e}=\left[u_{i}\;v_{i}\;w_{i}\;\theta_{xi}\;\theta_{yi}\right].$$

The four-node quadrilateral element, referred to as iQS4 [24], is a C^0^-continuous, anisoparametric interpolation element with six DOF per node,
(8)$$u_{i}^{e}=\left[u_{i}\;v_{i}\;w_{i}\;\theta_{xi}\;\theta_{yi}\;\theta_{zi}\right],$$
where the third rotation variable, $\theta_{z}$, is a drilling DOF that represents the rotation about the z-axis. The details of the displacement interpolations and the corresponding shape functions for both iMIN3 and iQS4 are summarized in Appendix A and Appendix B, respectively.

Using Equation (3), the membrane and curvature strain measures that correspond to the experimental measurements can be obtained as
(9)$$e_{i}^{\varepsilon }={\left\{\begin{array}{l} \varepsilon_{x0}^{\varepsilon }\\ \varepsilon_{y0}^{\varepsilon }\\ \gamma_{xy0}^{\varepsilon } \end{array}\right\}}_{i}=\frac{1}{2}\left({\left\{\begin{array}{l} \varepsilon_{xx}^{+}\\ \varepsilon_{yy}^{+}\\ \gamma_{xy}^{+} \end{array}\right\}}_{i}+{\left\{\begin{array}{l} \varepsilon_{xx}^{-}\\ \varepsilon_{yy}^{-}\\ \gamma_{xy}^{-} \end{array}\right\}}_{i}\right)\;,\;i=1,\ldots ,N$$
(10)$$k_{i}^{\varepsilon }={\left\{\begin{array}{l} \kappa_{x0}^{\varepsilon }\\ \kappa_{y0}^{\varepsilon }\\ \kappa_{xy0}^{\varepsilon } \end{array}\right\}}_{i}=\frac{1}{2t}\left({\left\{\begin{array}{l} \varepsilon_{xx}^{+}\\ \varepsilon_{yy}^{+}\\ \gamma_{xy}^{+} \end{array}\right\}}_{i}-{\left\{\begin{array}{l} \varepsilon_{xx}^{-}\\ \varepsilon_{yy}^{-}\\ \gamma_{xy}^{-} \end{array}\right\}}_{i}\right)\;,\;i=1,\ldots ,N,$$
where ${\left\{\varepsilon_{xx}^{+}\;\varepsilon_{yy}^{+}\;\gamma_{xy}^{+}\right\}}^{T}$ and ${\left\{\varepsilon_{xx}^{-}\;\varepsilon_{yy}^{-}\;\gamma_{xy}^{-}\right\}}^{T}$, denote the in-plane normal and shear strains measured on the top (z=t) and bottom (z=−t) surfaces at $\left(x_{i},y_{i}\right)$ locations, respectively, 2t refers to the thickness of the shell, and N refers to the total number of strain–sensor locations.

The iFEM methodology is based on an error functional that requires the differences between the analytical and experimental strain measures to be minimized in a least-squares sense. For each inverse finite element, e, this functional can be written as
(11)$$\Phi_{e}(u^{e})=w_{e}||e(u^{e})-e^{\varepsilon }{\left|\right|}^{2}+w_{k}||k(u^{e})-k^{\varepsilon }{\left|\right|}^{2}+w_{g}||g(u^{e})-g^{\varepsilon }{\left|\right|}^{2},$$
where $w_{e},\;w_{k}$, and $w_{g}$ are row vectors of weighting coefficients that control the degree of influence of the experimental strain measures to those described analytically. Vectors $w_{e}$ and $w_{k}$ are associated with the three in-plane membrane strain components and three bending curvatures, respectively, while $w_{g}$ corresponds to the two transverse shear strain components. Numerous potential scenarios can be handled by choosing an appropriate value for these coefficients. Some of these cases are discussed below.

In an element where the experimental strain measures are known, the weighting coefficients can be set to unity $\left(w_{e}=w_{k}=\left\{1,\;1,\;1\right\}\right)$, and the corresponding squared norms for the element can be calculated as
(12)$$\begin{array}{l} ||e(u^{e})-e^{\varepsilon }{\left|\right|}^{2}=\frac{1}{A_{e}}\int_{A_{e}}{\left[e\left(u^{e}\right)-e^{\varepsilon }\right]}^{2}dA\;,\\ ||k(u^{e})-k^{\varepsilon }{\left|\right|}^{2}=\frac{{\left(2t\right)}^{2}}{A_{e}}\int_{A_{e}}{\left[k\left(u^{e}\right)-k^{\varepsilon }\right]}^{2}dA\;, \end{array}$$
where $A_{e}$ is the area of the inverse element.

In cases where experimental data are not available and the experimental strain measures cannot be calculated, the weighting coefficients are set to a suitable small value and the corresponding squared norms are given as
(13)$$\begin{array}{l} ||e(u^{e})-e^{\varepsilon }{\left|\right|}^{2}=\frac{1}{A_{e}}\int_{A_{e}}e{\left(u^{e}\right)}^{2}dA\;,\\ ||k(u^{e})-k^{\varepsilon }{\left|\right|}^{2}=\frac{{\left(2t\right)}^{2}}{A_{e}}\int_{A_{e}}k{\left(u^{e}\right)}^{2}dA\;. \end{array}$$

The use of a small value for the weighting coefficient reduces the contribution of the elements that do not have the strain measurements to the global error functional. The inter-element kinematic compatibility is ensured across the entire iFEM mesh regardless of whether or not the inverse elements have measured strain data. Since the transverse shear strain measures, $g^{\varepsilon }$, cannot be obtained directly from experimental strains, the corresponding weighting coefficient is always set to a small value $\left(w_{g}=\left\{10^{-4},\;10^{-4}\right\}\right)$ and the squared norms for all elements can be computed according to the expression:(14)$$||g(u^{e})-g^{\varepsilon }{\left|\right|}^{2}=\frac{1}{A_{e}}\int_{A_{e}}g{\left(u^{e}\right)}^{2}dA\;.$$

Alternatively, Equation (14) can be used with $w_{g}=\left\{1,\;1\right\}$, leading to a strong enforcement of the Kirchhoff (zero transverse shear) constraints. This would be fully consistent for application to thin plates, where the transverse shear deformations are much smaller than those due to the bending deformations. For application to moderately thick plates, a post-processing procedure can be employed to obtain the $g^{\varepsilon }$ strains from the transverse shear equilibrium equations, using an a priori applied smoothing technique [22]. If such an approach is undertaken, the $g^{\varepsilon }$ contribution would be fully included in Equation (14), as in Equation (12), while using large weighting coefficients, i.e., $w_{g}=\left\{1,\;1\right\}$.

Since the squared norms described above involve area integrals, a suitable numerical integration scheme, e.g., Gauss quadrature, can be used. Depending on the Gauss scheme chosen, the experimental strain measures must be evaluated at each Gauss point of the element. In cases where there is an unavailability of strain measurements at certain Gauss points, a suitable weighting coefficient can also be associated with each point and a suitable low value can be provided as described above [32].

A set of linear algebraic equations is obtained by minimizing the error functional in Equation (11) with respect to the nodal degrees of freedom,
(15)$$\frac{\partial \Phi_{e}(u^{e})}{\partial u^{e}}=k^{e}u^{e}-f^{e}=0\Rightarrow k^{e}u^{e}=f^{e},$$
where the square matrix, $k^{e}$, is a function of the strain–sensor positions and the vector, $f^{e}$, is a function of the measured strain data. These quantities are analogous to the element stiffness matrix and force vector obtained in a direct FE analysis. Both $k^{e}$ and $f^{e}$ can be expanded and written in terms of the derivatives of the element shape functions,
(16)$$\begin{array}{l} k^{e}=\frac{1}{A_{e}}\int_{A_{e}}\left[w_{e}{\left(B^{m}\right)}^{T}B^{m}+w_{k}{\left(2t\right)}^{2}{\left(B^{b}\right)}^{T}B^{b}+w_{g}{\left(B^{s}\right)}^{T}B^{s}\right]\;dA,\\ f^{e}=\frac{1}{A_{e}}\int_{A_{e}}\left[w_{e}{\left(B^{m}\right)}^{T}e^{\varepsilon }+w_{k}{\left(2t\right)}^{2}{\left(B^{b}\right)}^{T}k^{\varepsilon }+w_{g}{\left(B^{s}\right)}^{T}g^{\varepsilon }\right]\;dA. \end{array}$$

The local contributions from all the elements can be assembled using an appropriate element transformation matrix, $T^{e}$, to transform the local element coordinate system to the global coordinate system of the structure (refer to Appendix C). The global matrices and vectors can be assembled as
(17)$$K=\sum^{N_{e}}{\left(T^{e}\right)}^{T}k^{e}T^{e}\;,\;F=\sum^{N_{e}}{\left(T^{e}\right)}^{T}f^{e}\;,\;U=\sum^{N_{e}}{\left(T^{e}\right)}^{T}u^{e}\;,$$
where $N_{e}$ denotes the total number of inverse Finite Elements. Finally, the following global set of linear algebraic equations can be solved to obtain the nodal displacements, U, of the structure,
(18)KU=F.

The solution of Equation (18) involves the application of the requisite displacement boundary conditions to restrain the structure against rigid-body motion. Subsequently, provided **K** is non-singular, the DOF vector, **U**, can be uniquely determined. Note that, in addition to specifying the displacement boundary conditions, the number of strain sensors and their distributions across the iFEM mesh are key factors to achieving a non-singular **K** matrix.

It is emphasized herein that the present iFEM formulation is based on the strain–displacement relations, and, thus, it is independent of the material properties and loading conditions of the structure. This inverse formulation can also be used regardless of the initial state of the structure, e.g., in pre-stressed structures. Although the presence of a membrane pre-stress affects the direct FE solution, due to the stiffening of the structure, it does not alter the strain–displacement relations used in iFEM. When such a structure is analyzed using iFEM, the initial structural geometry would include the pre-stressed displacement field. Any additional strains and resulting displacements would be related by the strain–displacement relations; hence, iFEM would reconstruct the latter using the former as input. A flowchart describing the various steps involved in implementing the iFEM procedure is shown in Figure 2.

## 3. Numerical Results

A numerical study is performed to ascertain the effects of using a sparse set of strain–sensor data on the iFEM accuracy. It is expected that reduction in the number of sensors would result in lower reconstruction accuracy. Quantifying this change will help in a more objective assessment of any given strain–sensor configuration. For practical iFEM applications, it is important to design potentially suitable strain–sensor configurations that would yield stable and accurate reconstruction of the deformed structural shape.

### 3.1. Problem Definition

The study is focused on the displacement reconstruction (shape sensing) of various vibration modes of a cantilevered Aluminium rectangular plate. The plate is clamped at one of its short ends, and has the following dimensions and material properties: length, a = 3 m, width, b = 1 m, thickness, 2t = 1 mm, Young’s modulus, E = 73 GPa, Poisson’s ratio, ν = 0.3 and material density, ρ = 2700 Kg/m^3^ (refer to Figure 3).

The proposal of using low- and high-frequency vibration modes for the iFEM-based shape-sensing analysis stems from the idea that the individual modes represent a varying degree of complexity of structural response. Thus, low-frequency modes have relatively simple shapes, whereas high-frequency modes generally involve more complicated deformed shapes and strain/stress distributions. Table 1 summarizes the select set of vibration modes considered in this study. Note that linear combinations of such modes can also be used to represent more general loading conditions.

A high-fidelity FE model of the plate is developed in ABAQUS using the S4R element, which is a four-node shell element based on Mindlin plate theory, with a bi-linear displacement field and reduced integration (1-point Gauss quadrature) of the transverse shear strain energy. The FE model used a high-density mesh of 4800 elements. The FE results serve as the reference displacement field and provide the simulated experimental strain data required for iFEM. Two types of inverse elements are used: a three-node element (iMIN3) and a four-node element (iQS4). Although the element formulations are developed for general shell applications, the example problems considered in this study address only linear bending and twisting of the plate according to Mindlin theory (transverse shear deformation included). Hence, the in-plane kinematic variables are identically zero and do not contribute to the bending and twisting deformations. As the in-plane strain distribution across the plate thickness is anti-symmetric with respect to the mid-plane, only strain–sensors positioned either on the plate’s top or bottom surface are required to compute the experimental strain measures. This study used strain–sensors placed only on the top surface of the plate.

Throughout this study, several strain–sensor schemes will be examined, with the aim of achieving relatively small number of strain sensors that guarantee accurate and stable iFEM reconstruction solutions. For those inverse finite elements that are “instrumented” with strain sensors, only a single strain sensor will be used, thus representing a uniform (constant) distribution of the measured strains for these elements. Since iMIN3 has constant membrane and bending strain measures, it is sufficient to integrate the corresponding error norms using a 1-point Gauss quadrature numerical integration. The full integration of the error norms due to the transverse shear deformations requires a 3-point Gauss quadrature since the transverse shear strains are linear across the element domain. For this element, however, the 1-point Gauss-quadrature integration of the transverse shear terms is both sufficient and accurate. The iQS4 element interpolates u, v, and w bi-quadratically, whereas the $\theta_{x}$ and $\theta_{y}$ rotations are bi-linear, thus requiring a 3 × 3 Gauss to achieve exact numerical integration of the error norms. From the perspective of computational efficiency, a 2 × 2 Gauss integration is sufficiently accurate. Note that the FE reference models will be used to extract the simulated “experimental” strains at the element centroids, simulating the strain sensor values.

Three different sparse-sensor configurations are investigated (see Figure 4). Configuration A has strain sensors placed only along the centroids of the boundary elements, Configuration B is designed by placing additional interior sensors, diagonally, to form a zig-zag pattern and, finally, Configuration C supplements the previous two configurations with sensors placed along both cross-diagonal paths. Although the primary aim of Configuration A is to minimize the number of sensors required and accurately capture the strains along the boundaries, Configurations B and C are inspired by the need for more internal sensor data; these sensor patterns bear a resemblance to the designs used for industrial load bearing frame structures.

It is important to remark that the proposed strain–sensor patterns have a common design feature. All three patterns have continuous strain–sensor distributions, i.e., adjacent inverse elements in a strain–sensor pattern have strain data. Numerical studies have determined that, when discontinuous strain–sensor distributions are used, Equation (17) can result in a singular system matrix, **K**, hence no iFEM solution is attainable. Similarly, the strain–sensor patterns with an insufficient number of boundary elements populated with strain data can also result in a singular **K** matrix. This is especially true for boundary elements that have nodes where the displacement boundary conditions are prescribed. Hence, in the strain–sensor patterns examined herein, all boundary elements are instrumented with strain sensors.

The initial set of results are obtained using the iQS4 element. The iQS4 element mesh has a total of 1200 elements, with 60 elements along the length and 20 along the width of the plate. The iMIN3 element is also used, based on the quadrilateral element mesh, so that comparisons between the iMIN3 and iQS4 reconstructions can be made.

### 3.2. Reconstruction of the First Two Mode Shapes

The first two vibrational modes of the plate correspond to the frequencies, 0.094 and 0.58 Hz, and represent the first bending and torsional modes, respectively. The first mode represents a simple case of plate bending, where the displacement components *w* and *θ_y_* are dominant. The second mode represents a case of plate twist, where the *θ_x_* rotation is dominant. Taken together, these two cases explore all relevant deformation avenues of the plate and will help establish a baseline performance of iFEM when using sparse sensor data.

For the initial reconstruction, a full-field set of sensor data is used, i.e., experimental strain data is measured at the centroids of all iQS4 elements (referred to as Configuration D). The contour plot of the iFEM reconstructed deflection, *w*, is shown in Figure 5. Comparison of the reconstructed results with reference FE results indicate high accuracy, with the maximum percent error in the deflection of less than 0.02%. A similar level of accuracy is observed for the *θ_x_* and *θ_y_* rotations. Due to the high density of sensor data, the results of the present case represent the highest accuracy that the iFEM model can offer, and it serves as the iFEM reference solution for the reconstructions based on sparse sensor data. Although the contour plot of the results, as seen in Figure 5, is instructive and shows a spatial variation of the deflection field, it is difficult to assess any specific details in the reconstructed shape. Henceforth, line plots are presented, taken along various sections of the plate that capture the maximum in the various deformation components for each mode. Each line plot is normalized using a normalization factor equal to the maximum of the reference FE results along that section.

The reconstruction results using Configurations A, B, and C, are plotted for Mode 1 and compared with the reference FE results. The line plots of the deflection, *w*, and rotation, *θ_y_*, along sections A-A′ and B-B′ (see Figure 5b for the section definition), for Mode 1 are shown in Figure 6 and Figure 7. The plots indicate that the iFEM results for Mode 1 show remarkable similarity with the FE results, with a maximum error of less than 0.2% for *w*, and less than 0.53% for *θ_y_* among the configurations. These reconstruction results are extremely accurate as they show an error of less than 1% across all the sparse configurations.

Since Mode 1 is the first bending mode, the effect of rotation, *θ_x_*, is minimal. Hence, Mode 2 (the first torsional mode) is also explored. The contour plot of the deflection, *w*, using a full-field set of sensors is shown in Figure 8. As in the previous case, these results are highly accurate, with a maximum error of less than 0.03%.

As seen for the case of Mode 1, these reconstruction results are focused on the deflection, *w*, and rotation, *θ_x_*, as they are the dominant kinematic variables for this mode. iFEM reconstruction is again performed using the sparse Configurations A, B, and C, and the results are shown as line plots along sections C-C′ and B-B′ (see Figure 5b and Figure 8b for the section definitions). The results shown in Figure 9 and Figure 10 indicate high reconstruction accuracy with a maximum error of less than 0.25% for *w* and 0.76% for *θ_x_*. The level of accuracy is again similar to what was observed for Mode 1.

These results indicate that all three sparse sensor configurations are equally suited to be used for the reconstruction of the first two vibrational modes. The first two modes are examples of relatively simple transverse deflection and rotation fields, and these results are also highly accurate for all three deformation components. As the purpose of the present effort is to quantify the performance limits of these strain–sensor configurations, the results for the first two modes can be used to define the lower limits of performance.

To define the upper limits of performance, more complex displacement fields need to be examined. For this purpose, iFEM is used to reconstruct several higher mode shapes of the plate.

### 3.3. Reconstruction of Higher Mode Shapes

For demonstration purposes, Mode 6 is now considered, which is the fourth bending mode of the plate, corresponding to a frequency of 3.27 Hz. The deflection shape has a total of four nodes and anti-nodes and presents a relatively more complex displacement field suitable for establishing an upper limit of reconstruction performance. The contour plot of the deflection, *w*, using a full-field set of sensors, is shown in Figure 11. The results again are highly accurate, with a maximum error in *w* of 1.3%, measured at the tip of the plate. The reconstruction of the rotation components is of similar accuracy. Hence, the iFEM results of Mode 6 for Configuration D indicate a level of accuracy similar to the lower modes.

Next, the strain data corresponding to the sparse Configurations A, B and C are used for iFEM reconstruction. Figure 12 depicts the deflection distributions along sections A-A′ and B-B′. These results clearly show significant variations in performance among the three configurations. The Configuration A results show a maximum error of 10.54%, with the location of the errors occurring at the internal local deflection peaks of the mode shape. The deflection is more accurate at the two outer boundaries of the plate, with a maximum error of 0.75%. The results of Configurations B and C offer greater accuracy with errors less than 1.5% throughout the plate.

A similar level of accuracy was observed for rotation, *θ_y_* (see Figure 13), with a maximum error of 8.37% for Configuration A, whereas the other two configurations are more accurate with errors less than 1.5%. Figure 14 shows the reconstruction results for rotation, *θ_x_*, with results of Configurations A, B, and C showing maximum errors of 18.14%, 4.95% and 3.43%, respectively. Here, Section D-D′ (see Figure 8b for the section definition) is used for the plot, as it intersects a node with the global maximum in *θ_x_* for Mode 6.

Mode 6 results of Configuration A show that the main inaccuracies are present at internal nodes of the plate. This is attributed to the fact that, in Configuration A, sensors are present only along the boundaries. Hence, there is a lack of strain information within the plate interior, leading to difficulties in reconstructing internal displacements while the boundary displacements are accurate. This is further corroborated by the results of Configurations B and C, where the presence of internal sensors leads to greater accuracy within the plate. The effect of reconstruction accuracy along sections with a sparse and dense set of sensor data is further explored. This can be understood by considering sections C-C′ and B-B′, both of which lie along elements with strain data. The line plots along these sections are shown in Figure 15. These results clearly indicate that the displacement and rotation reconstruction along these paths are highly accurate, with a maximum error of 1.24% for *w* and 0.70% for *θ_x_* among the configurations. These results are in direct contrast to those seen previously, as they show that, even for Configuration A, highly accurate results are obtained along densely populated sensor paths for Mode 6.

A similar case is explored by considering the results along paths G-G′ and H-H′ (see Figure 11b for the section definition), which correspond to the cross-diagonal sensor patterns of Configuration C. These results will also help to discriminate between Configurations B and C, which until now offered comparable reconstruction accuracies. These plots are shown in Figure 16 and Figure 17. The results clearly show that Configuration A is less accurate. However, further scrutiny reveals the differences between Configurations B and C. They show that Configuration B has a maximum error of 2.52% for w and 9.47% for $\theta_{x}$, whereas Configuration C has a maximum error of less than 1% for the two variables.

These plots also indicate asymmetries in the results of Configuration B. Along section G-G′, $\theta_{x}$ has a maximum error of 9.47%, whereas path H-H′ shows a maximum error of 1.05%. This asymmetry is attributed to the fact that the former path only has strain sensors in the boundary elements, whereas the latter path has strain sensors all along the path. This again reinforces the conclusion that deflection reconstruction is more accurate along a path with a dense set of sensors. It also demonstrates the greater accuracy of using a symmetric cross-diagonal pattern similar to that used in Configuration C. The current set of results help demonstrate that Configuration C is the most accurate sensor pattern across all the explored cases. For practical applications, however, Configuration B could be a more optimal choice, as it offers a great trade-off between reconstruction accuracy and the number of required sensors.

### 3.4. Comparing Results of iMIN3 and iQS4 Models

The results presented until now were obtained using the iQS4 element. In this section, these results are compared with results obtained using the iMIN3 element for the same problem cases explored above. Such a comparison would prove to be an interesting investigation into the effect of the choice of inverse element on the reconstruction performance of a specific problem. For each sensor configuration, the normalized displacement components are plotted as a function of the number of sensors used to show the convergence of the results. The results from both the sparse and full-field sensor configurations are used for this study.

For a parametrically equivalent comparison between the two sets of elements, it is necessary to use an identical set of sensor data for both inverse element cases. To achieve this equivalence, the iMIN3 mesh is derived from that used for the iQS4 element by dividing each quadrilateral element cross diagonally to get four triangular elements, as shown in Figure 18. In addition, the strain data measured at the centroid of each quadrilateral element are used equally for all four triangular elements occupying that area. Using such a strategy, equivalent sparse sensor configurations are obtained using both iMIN3 and iQS4 elements. The details on the number of strain rosettes used are summarized in Table 2.

iFEM reconstruction results using Configurations A, B, C, and D are plotted for Modes 5 and 6. The contour plot of iFEM reconstructed deflection, *w*, for Mode 5 using a full-field set of sensors is shown in Figure 19. For each mode, the results are plotted at specific locations of the plate corresponding to locations of maximums in $w,\;\theta_{x}$ and $\theta_{y}$. For Mode 5 (second torsional mode), these nodes are located at coordinates: (2.95, 0.95), (2.95, 0.50), and (2.00, 0.95), respectively. Similarly, the points chosen for the Mode 6 plots are: (2.35, 0.50), (2.50, 0.90), and (2.00, 0.50). The convergence results for Mode 5, plotted using a logarithmic scale along the *x*-axis, are shown in Figure 20.

These results indicate that the kinematic variables $w,\;\theta_{x},$ and $\theta_{y}$ converge to FE results with an increasing number of sensors used. The plots also indicate similar predictive capability of the iMIN3 and iQS4 results. The convergence results for Mode 6 are shown in Figure 21, and similar conclusions are drawn from these plots.

The numerical study discussed above demonstrated the performance of three different sparse sensor Configurations A, B, and C for reconstructing various simple and complex displacement fields. The configurations were chosen to illustrate increasing levels of complexity in sensor patterns, suitable for practical applications. The results clearly demonstrate that Configuration A, with its fully defined boundary sensor data, is suitable for accurately reconstructing simple displacement fields as seen for Modes 1 and 2. However, more complex displacement fields required the use of internal strain data as was achieved by Configurations B and C. Reconstruction asymmetries were observed for Configuration B, whereas Configuration C consistently produced accurate results. Finally, Configuration B was identified as an ideal choice for practical applications, as it offered a good compromise between the number of sensors used and reconstruction accuracy. In addition, an investigation into the reconstruction using two different inverse elements revealed similar results. This indicates that, irrespective of the inverse element used, similar performance can be expected when using the same set of experimental strain data.

Although the present study was shown to be highly effective, the proposed sparse sensor configurations involve relatively dense sensor patterns. A number of strain–sensor reduction strategies can be used to reduce the number of strain sensors even further. For example, when fewer strain sensors are available along a one-dimensional sensor pattern while a high-density inverse element mesh is employed, the strain values in the elements without strain data can be obtained by way of simple interpolation or smoothing methods. However, another strategy is to make use of fiber-optic strain sensors, permitting the desired high-strain measurement density. Note that the proposed configurations involve long one-dimensional sensor paths that can easily be instrumented using fiber optic sensors. Moreover, the use of a relatively coarse iFEM mesh would necessarily allow for a much smaller number of strain sensors, although less accurate reconstruction results would be expected as a result.

## 4. Conclusions

This paper presented a detailed investigation into the effects of sparse strain–sensor data on the displacement reconstruction using the inverse FEM (iFEM). The iFEM discretizes a structural domain by means of inverse finite elements that are formulated on the basis of a weighted-least-square error functional being computed. The functional represents the least-square error between the experimentally measured strains and those corresponding to the finite element approximations. Suitable values of weighting coefficients are used to account for the presence or absence of strain data within each element. In elements with strain data, the weighting coefficients are equated to unity, whereas, in elements without strain data, the weighting coefficients are set to be several orders of magnitude smaller, thus reducing the element contribution towards the global error functional. The use of these weighting coefficients allows the use of iFEM for problems with sparse strain–sensor data.

Both three and four-node inverse elements were used in the study of a cantilevered rectangular plate undergoing low and moderately high frequency vibrational motion. The elements were derived from the displacement assumptions of Mindlin theory, using $C^{0}$-continuous anisoparametric shape functions adopted from plate formulations of the direct FEM. Inspired by the shape of load-bearing frame structures, three sparse sensor configurations were explored in the computational study. These sensor configurations were used for the shape sensing of low and higher bending and torsional vibration modes. These modes encompass both simple and complex displacement fields, where all the major kinematic variables were excited.

The iFEM performance using the sparse sensor configurations was assessed by comparing the reconstruction results with the reference FE results. For the first bending and torsional mode, highly accurate displacements and rotations were obtained using all three sparse sensor configurations. However, for the fourth bending mode, sensors placed only along the boundaries were insufficient to reconstruct the complex internal displacement distribution and required additional strain–sensor data, specifically in the interior of the plate. These results highlighted the effectiveness of only using boundary strain data for reconstructing relatively simple displacement fields, whereas superior reconstructions can be achieved by using additional internal sensors along zig-zag or cross-diagonal paths. It was also inferred that, for any sensor configuration, displacement reconstruction along a path with a dense sensor array would be more accurate than along a path with very few sensors. This indicates that reconstruction accuracy decreases with increasing distance from strain sensors. In addition, an asymmetric sensor pattern could lead to asymmetries in the reconstruction results, as observed for the zig-zag pattern. Although the cross-diagonal sensor patterns provided the most accurate results, the zig-zag sensor pattern presents a reasonable practical compromise, offering high reconstruction accuracy with low sensor data requirements. Both the three- and four-node inverse element models were also used to study solution convergence of the kinematic variables, with increasing sensor density. Excellent correlations with reference FE results were observed, and both element types produced comparable high-accuracy results.

Although the sensor configurations explored in this work have been demonstrated to be effective for a simple rectangular plate, they are intended to inspire further designs for applications involving more complex structures. The concepts of using a continuous line of sensors along the boundary and using cross-diagonal paths can be amalgamated to create novel configurations satisfying the application requirements. Such an investigation involving more geometrically complex structures is identified as a topic for future work. Additionally, while the current work focused on problems involving strain rosettes, the relative simplicity of the proposed patterns offers the possibility of extending the present work to strain measurements using fiber-optic sensors. The greater number of strain measurements using fiber-optics could help supplement the level of accuracy already observed in the current set of results and is also a topic for future investigations.

## Figures and Tables

**Figure 1 sensors-20-07049-f001:**
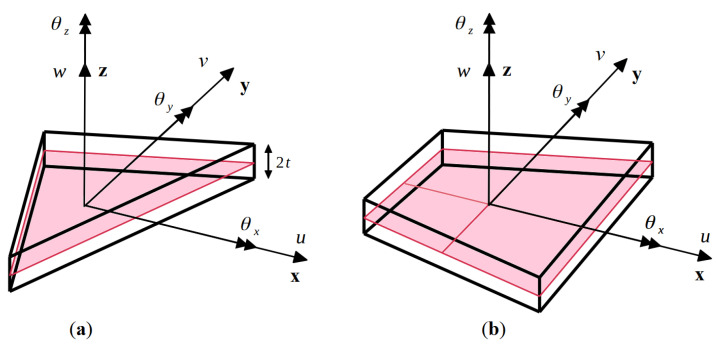
Sign conventions for the kinematic variables of the two inverse elements: (**a**) three-node triangular element, iMIN3; and (**b**) four-node quadrilateral element, iQS4.

**Figure 2 sensors-20-07049-f002:**
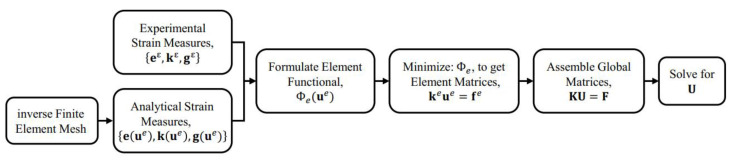
Flow chart of the iFEM procedure.

**Figure 3 sensors-20-07049-f003:**
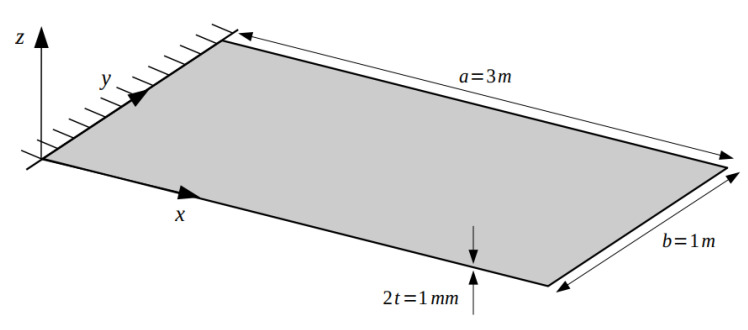
Geometrical dimensions and boundary conditions of the rectangular plate used for the numerical study.

**Figure 4 sensors-20-07049-f004:**
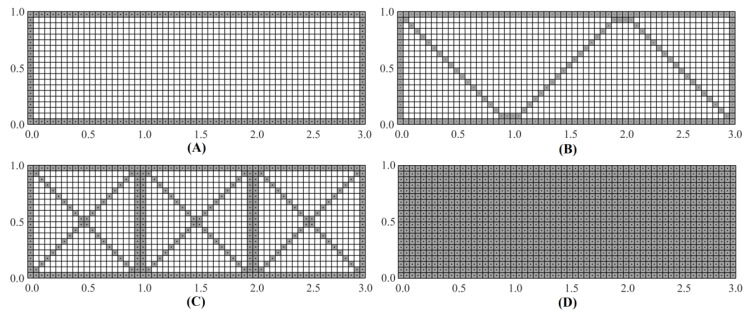
Sensor configurations for the iQS4 mesh; Configuration: (**A**) with strain sensors along the boundaries; (**B**) with strain sensor along diagonal paths arranged in a zig-zag pattern; (**C**) with sensors along cross-diagonal paths; and (**D**) with sensors placed at the centroids of all elements.

**Figure 5 sensors-20-07049-f005:**
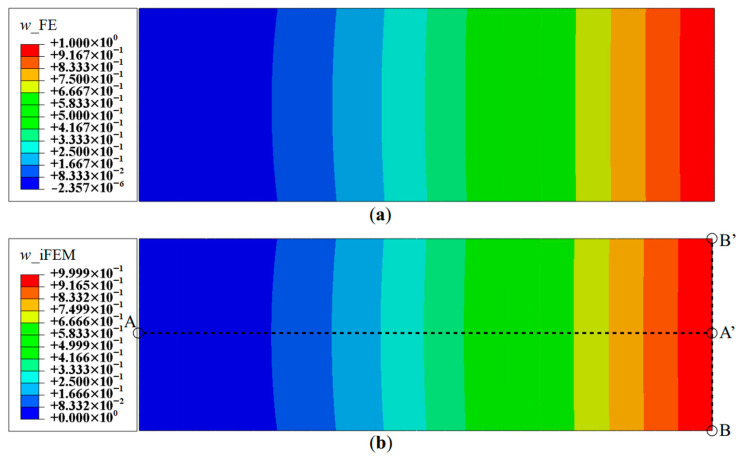
Contour plots of deflection for Mode 1: (**a**) FE results; and (**b**) iFEM results using full-field strain data (Configuration D).

**Figure 6 sensors-20-07049-f006:**
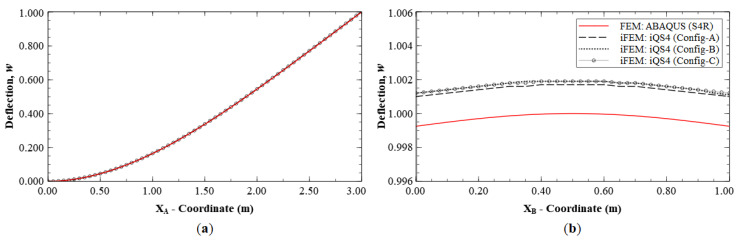
Line plots of deflection (*w*) for Mode 1 along: (**a**) section A-A′; and (**b**) section B-B′.

**Figure 7 sensors-20-07049-f007:**
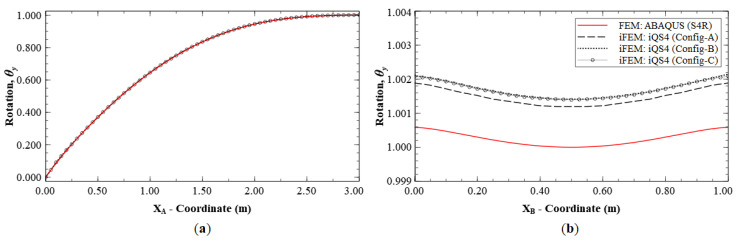
Line plots of rotation ($\theta_{y}$) for Mode 1 along: (**a**) section A-A′; and (**b**) section B-B′.

**Figure 8 sensors-20-07049-f008:**
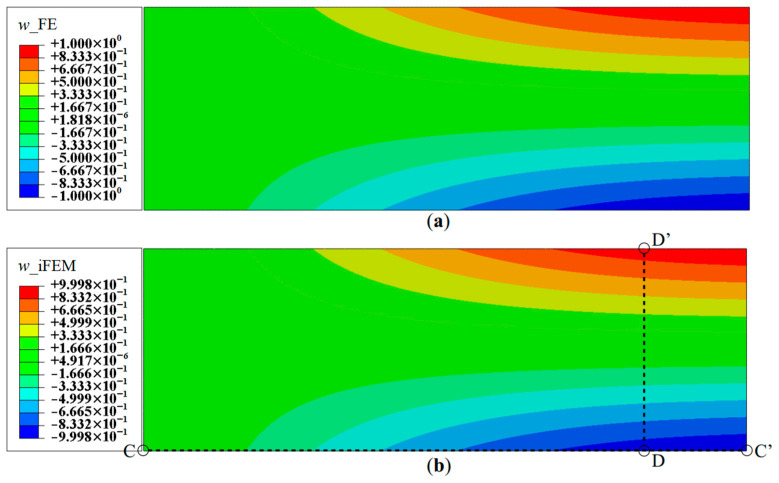
Contour plots of deflection for Mode 2: (**a**) FE results; and (**b**) iFEM results using full-field strain data (Configuration D).

**Figure 9 sensors-20-07049-f009:**
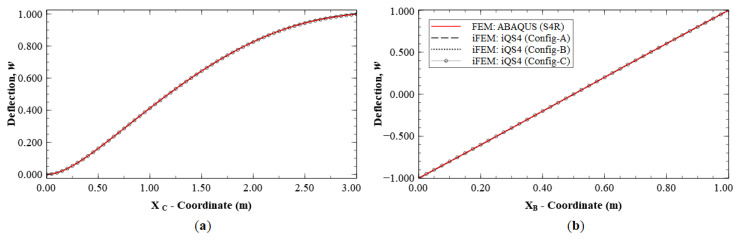
Line plots of deflection (*w*) for Mode 2 along: (**a**) section C-C′; and (**b**) section B-B′.

**Figure 10 sensors-20-07049-f010:**
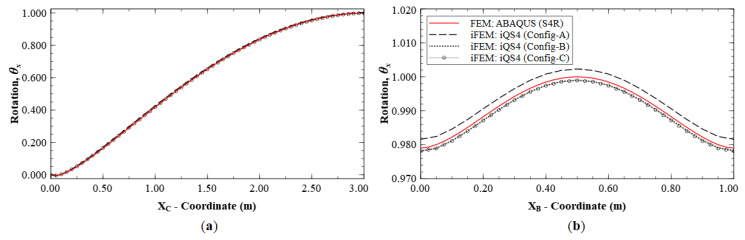
Line plots for rotation ($\theta_{x}$) for Mode 2 along; (**a**) section C-C′; and (**b**) section B-B′.

**Figure 11 sensors-20-07049-f011:**
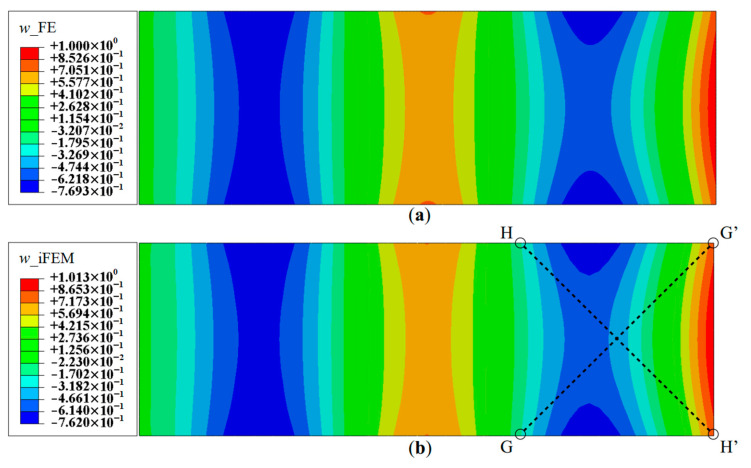
Contour plots of deflection for Mode 6: (**a**) FE results; and (**b**) iFEM results using full-field strain data (Configuration-D).

**Figure 12 sensors-20-07049-f012:**
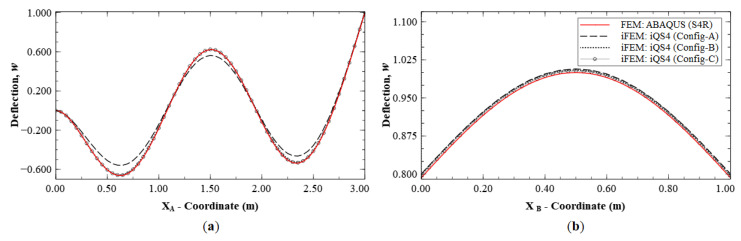
Line plots of deflection (*w*) for Mode 6 along: (**a**) section A-A′; and (**b**) section B-B′; (see Figure 5b for the section definitions).

**Figure 13 sensors-20-07049-f013:**
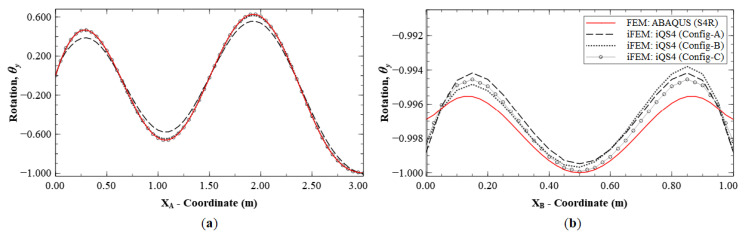
Line plots of rotation ($\theta_{y}$) for Mode 6 along; (**a**) section A-A′; and (**b**) section B-B′; (see Figure 5b for the section definition).

**Figure 14 sensors-20-07049-f014:**
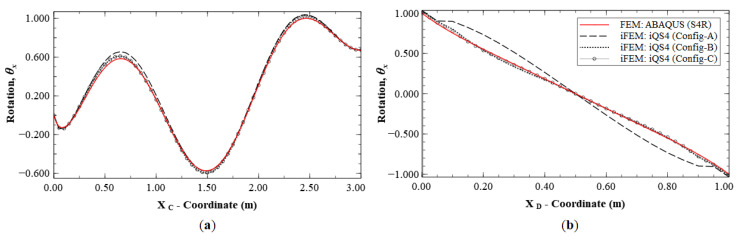
Line plots of rotation ($\theta_{x}$) for Mode 6 along; (**a**) section C-C′; and (**b**) section D-D′; (see Figure 8b for the section definition).

**Figure 15 sensors-20-07049-f015:**
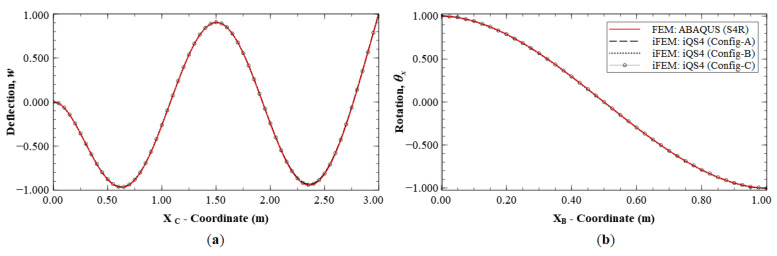
Reconstruction results for Mode 6; line plots of: (**a**) deflection *w*, along section C-C′; and (**b**) rotation $\theta_{x}$, along section B-B′; (see Figure 5b and Figure 8b for the section definitions).

**Figure 16 sensors-20-07049-f016:**
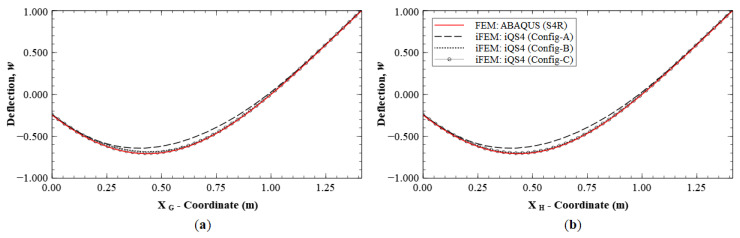
Line plots of deflection (*w*) for Mode 6 along: (**a**) section G-G′; and (**b**) section H-H′.

**Figure 17 sensors-20-07049-f017:**
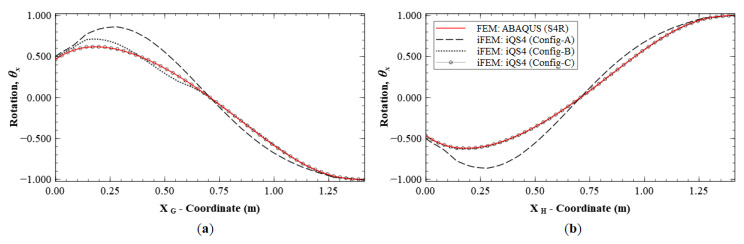
Line plots of rotation ($\theta_{x}$) for Mode 6 along: (**a**) section G-G′; and (**b**) section H-H′.

**Figure 18 sensors-20-07049-f018:**
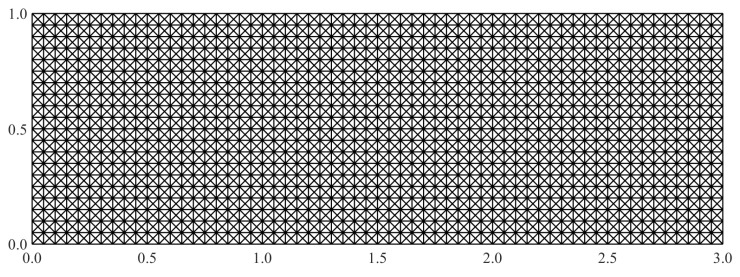
The iFEM mesh used for the iMIN3 element.

**Figure 19 sensors-20-07049-f019:**
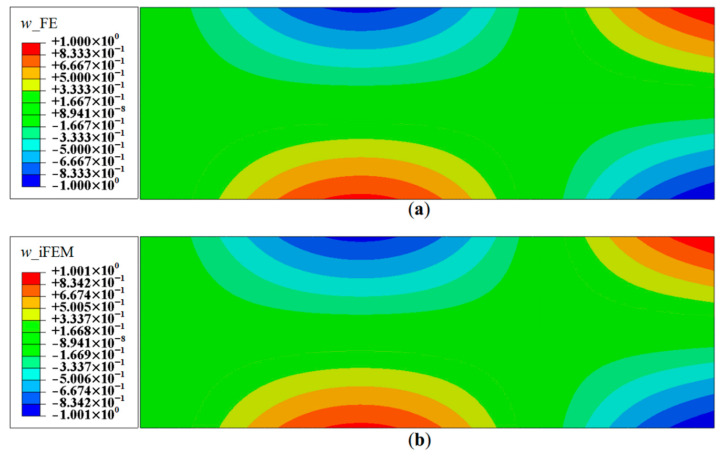
Contour plots of deflection for Mode 5: (**a**) FE results; and (**b**) iFEM results using full-field strain data (Configuration-D).

**Figure 20 sensors-20-07049-f020:**
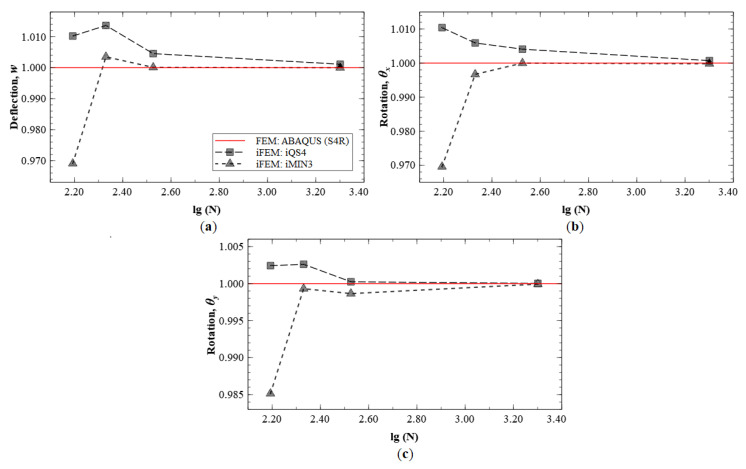
iFEM results for Mode 5: convergence plots for (**a**) *w*, (**b**) $\theta_{x}$, and (**c**) $\theta_{y}$.

**Figure 21 sensors-20-07049-f021:**
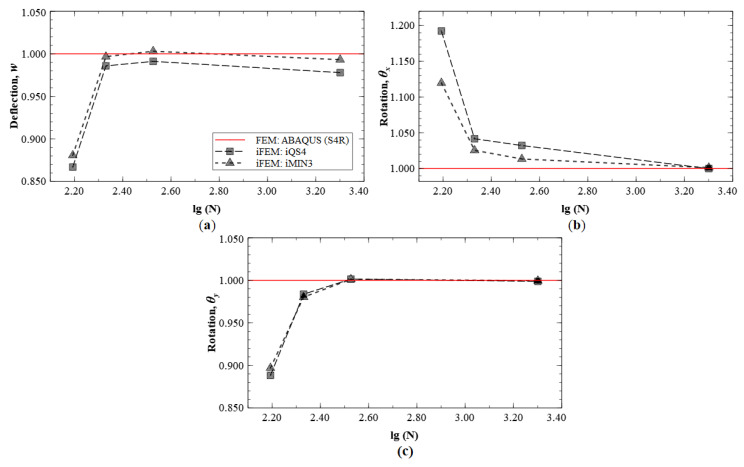
iFEM results for Mode 6: convergence plots for (**a**) *w*, (**b**) $\theta_{x}$, and (**c**) $\theta_{y}$.

**Table 1 sensors-20-07049-t001:** Description of the various vibration modes of the cantilevered plate.

Mode Number	Frequency	Mode Type
1	0.094 Hz	1st bending mode
2	0.58 Hz	1st torsional mode
5	1.84 Hz	2nd torsional mode
6	3.27 Hz	4th bending mode

**Table 2 sensors-20-07049-t002:** Strain data information for each sparse sensor configuration and inverse element.

Configuration	No. of Strain Rosettes Used	No. of Elements with Strain Data	
	N	iQS4 Mesh	iMIN3 Mesh
A	156	156	624
B	214	214	856
C	336	336	1344
D	1200	1200	4800

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
