# Peer review of "Shape Sensing of Plate Structures Using the Inverse Finite Element Method: Investigation of Efficient Strain–Sensor Patterns"

_sensors, 2020, doi:10.3390/s20247049_

Round 1
Reviewer 1 Report
This paper presents Shape sensing of plate structures using inverse Finite Element Method: Investigation of efficient strain sensor patterns. Before considering it for publication the authors are required to revise the paper based on the following remarks:
1- Please double check all the formulation (Bold, Italic, ….).
2- English typos should be revised.
3- The CPU time of each mesh should be mention to see the accuracy of the presented numerical model.
4- Mode types are missing (Number of modes belong to DOF).
5- Section 2 it will be better to describe the presented technique using a flowchart.
6- Please improve the Figs 11-13 using zoom. (The plotted results are very close but the error still higher? ) Please double-check carefully.
7- The introduction should be improved by different kinds of structures in dynamical analysis and inverse problem such as:
https://doi.org/10.1016/j.compstruct.2020.112497
https://doi.org/10.12989/scs.2020.34.4.511
Author Response
Please, see attachment

Reviewer 2 Report
See attached file

Author Response
Please, see the attachment

Round 2
Reviewer 1 Report
accept in this present form.